# Antibiotic Resistance Gene Transformation and Ultrastructural Alterations of Lettuce (*Lactuca sativa* L.) Resulting from Sulfadiazine Accumulation in Culture Solution

**Liang Mei, Ying-Xin Chen, Chao Wang, Jia-Hua Chen, Zhi-Jin Zhang, Min-Yao Zhou, Jin-Tao Feng and Yan Wang \***

College of Animal Science, South China Agricultural University, Guangzhou 510642, China; meiliang@stu.scau.edu.cn (L.M.); 19927537760@163.com (Y.-X.C.); WC20202122009@163.com (C.W.); 19927538670@163.com (J.-H.C.); z19927535779@163.com (Z.-J.Z.); zhouminyao00@163.com (M.-Y.Z.); fengjintaott@163.com (J.-T.F.)
\* Correspondence: ywang@scau.edu.cn

**Abstract:** The research herein explored the possible mechanism of toxicity of the antibiotic sulfadiazine (SD) and the related antibiotic resistance gene transformation in lettuce by systematically investigating its growth responses, ultrastructural changes, and antibiotic resistance gene transformation via solution culture experiments. The results showed that SD mainly accumulated in the roots of lettuce at concentrations ranging from 6.48 to 120.87 μg/kg, which were significantly higher than those in leaves (3.90 to 16.74 μg/kg). Lower concentrations of SD (0.5 and 2.0 mg/L) in the culture nutrient solution exerted little effect on lettuce growth, while at SD concentrations higher than 10 mg/L, the growth of lettuce was significantly inhibited, manifesting as shorter root length and lower dry matter yield of whole lettuce plants. Compared with that for the control group, the absolute abundance of bacteria in the root endophyte, rhizosphere, and phyllosphere communities under different concentrations of SD treatment decreased significantly. *sul1* and *sul2* mainly accumulated in the root endophyte community, at levels significantly higher than those in the leaf endophyte community. Studies of electrolyte leakage and ultrastructural characteristics of root and leaf cells indicated that lettuce grown in culture solutions with high SD concentrations suffered severe damage and disintegration of the cell walls of organs, especially chloroplasts, in leaves. Furthermore, the possible mechanism of SD toxicity in lettuce was confirmed to start with the roots, followed by a free flow of SD into the leaves to destroy the chloroplasts in the leaf cells, which ultimately reduced photosynthesis and decreased plant growth. Studies have shown that antibiotic residues have negative effects on the growth of lettuce and highlight a potential risk of the development and spread of antibiotic resistance in vegetable endophyte systems.

**Keywords:** sulfadiazine; lettuce; phytotoxic; hydroponic; antibiotic resistance gene

## 1. Introduction

Intensive animal farming is often associated with the use of considerable amounts of drugs, including antibiotics for disease prevention and animal growth [1,2]. Sulfadiazine [SD, 4-amino-N-(2-pyrimidinyl) benzene sulfonamide] is a potent antibacterial agent belonging to a large group of structurally related antibiotics: the sulfonamides. Attributed to its high efficacy and affordable price, SD has been long and widely used in intensive livestock production. Although the amounts of antibiotics excreted from animals vary with the type and dosage level of antibiotics, as well as the species and age of the animals [3], the overall majority (25–75%) of the administered drugs are excreted via feces and urine [4].

Due to its high nutrient content, animal manure is often applied as an organic fertilizer to arable lands for crop production, which has resulted in the large-scale introduction of antibiotics into the terrestrial environment. Furthermore, antibiotics could exert pressure on microorganisms, inducing antibiotic resistance, in the receiving environment. To date,

various kinds of antibiotics, including SD, have been widely detected at high concentrations ranging from 1.93 to 760 µg/kg in soils [5–7]. In recent years, the ecotoxicity of residual antibiotics and antibiotic resistance genes in soils have received increasing attention, and antibiotics have been confirmed to cause toxicity to a wide range of organisms, including soil microorganisms, earthworms, insects, and other endpoints [8–11]. With antibiotic contamination in soils, many planted edible crops have been reported to accumulate antibiotics in their organs, which could ultimately enter the human body through the food chain [12]. The occurrence of antibiotic resistance genes in soil, water, and livestock manure has been studied extensively [13–15]. In contrast, only limited studies are available regarding plant microbial resistance [16]. Lettuce may be consumed with little processing, and this may spread antibiotic resistance to humans through the food chain [16], posing health risks to humans.

Lettuce (*Lactuca sativa* L.) is a widely consumed vegetable worldwide, and it has a high germination rate, sensitivity to contaminants, and low genetic variability, making it economically and ecologically relevant for toxicity studies [17,18]. For example, Song reported changes in the antioxidative system of lettuce in response to arsenic toxicity [19]; Yadzi evaluated cadmium accumulation in lettuce cultivated in contaminated environments and found that several phytochelatin synthase genes contribute to the accumulation of cadmium [17]. Recently, antibiotics as pollutants and their toxicity in plants have received particular attention. Xu et al. reported that the phytotoxicity of sulfanilamide antibiotics on *Arabidopsis thaliana* generates both toxic effects and hormesis related to plant drug uptake [20]. Nasri et al. studied the impact of five antibiotics on phytotoxicity and genotoxicity in rice [9]. Antibiotic resistance has become a global problem; antibiotics are considered emerging environmental pollutants and have thus attracted extensive attention worldwide [21]. Recently, antibiotic resistance in various manure-fertilized vegetables, such as celery, pakchoi, and cucumber, has been reported [22]. As mentioned above, although research has been conducted on the effects of antibiotics on several plant species [23], the possible growth response mechanisms and ultrastructural alterations of leafy vegetables in response to antibiotics and whether the antibiotic resistance of endophytic bacteria can be directly impacted by antibiotic pollution in the environment are still unclear.

Therefore, in this study, the responses of *Lactuca sativa* L. organs, including roots and leaves, to SD and antibiotic resistance gene accumulation during the growth processes under SD stress in culture solutions were systematically studied to explore the possible mechanism of lettuce toxicity and antibiotic resistance gene selection induced by antibiotics. To achieve this goal, experiments were conducted to (i) investigate the quantitative responses of roots and leaves of *Lactuca sativa* L. to SD stress, (ii) determine SD and antibiotic resistance gene accumulation levels in different organs with different SD concentrations in cultures, and (iii) explore the extent of organ damage and ultrastructure alterations during the growth processes of *Lactuca sativa* L. under SD stress via the culture substrate. Our findings will facilitate a more accurate assessment of the potential risks posed by antibiotic contamination to food quality and environmental health.

## 2. Materials and Methods

Commercial seeds of lettuce (*Lactuca sativa* L.) were used as the source of plant materials in this study. Seeds were sterilized with a 6% sodium hypochlorite solution for 30 min, followed by eight rounds of vigorous washing in distilled water. Then, seeds were transferred into a temporary plastic germination box with sandy soil under continuous artificial lighting at 20 °C. The germinated seedlings were carefully dug up from the germination box. After being washed to remove sediments and soils from roots, the seeds were immediately transferred to containers filled with nutrient solution for a 10-day adaptation period to hydroponic conditions. The nutrient solution contained 2 mM $Ca(NO_3)_2$, 0.1 mM $KH_2PO_4$, 0.5 mM $MgSO_4$, 0.1 mM $KCl$, 0.7 mM $K_2SO_4$, 10 µM $H_3BO_3$, 0.5 µM $MnSO_4$, 0.5 µM $ZnSO_4$, 0.2 µM $CuSO_4$, 0.01 µM $Na_2MoO_4$, and 100 µM Fe(II)-EDTA, with pH values ranging from 5.5 to 6.0. On Day 11, the lettuce seedlings were carefully transferred and allocated to their

respective treatments (plastic containers 10 cm wide × 21 cm long × 10 cm deep). The SD (Sigma-Aldrich, St. Louis, MO, USA) concentrations in the culture nutrient solutions were set at 0.01, 0.05, 0.5, 2, 10, and 50 mg/L, with conditions without SD as the control. All experiments were conducted in triplicate, and all plants were maintained in an incubator at 20 °C.

The primary roots of the 0, 0.01, 0.05, 0.5, 2, 10, and 50 mg/L SD-treated plants were collected on Days 0, 7 and 14 and after the last measurement on Day 14. After first measuring the root length, all plants were carefully washed in distilled water and then individually separated into roots and leaves. Three roots from each treatment were randomly selected for the membrane permeability study. Plant samples from the control, 2 mg/L SD, and 10 mg/L SD treatments were selected to examine the cell structures via transmission electron microscopy (TEM). The lettuces in all the different treatments were harvested and weighed to study the dry matter yields and SD concentrations in lettuces after they were freeze-dried for 24 h.

A previously described sequential extraction procedure [24] was used for the extraction of SD with a minor modification. Briefly, 250 mg of leaf or root sample (in dry weight) was first extracted using 2 mL 0.01 M $CaCl_2$ as the mobile fraction for the total dissolved SD, followed by a second extraction using 10 mL methanol to extract the bioavailable SD fraction. The total SD concentrations in the organs were the sum of both of the above two extracted fractions. For each extraction step, samples were first sonicated for 30 min and then centrifuged for 30 min at $1700\times g$. The supernatants were mixed, evaporated, and dried by nitrogen stream to 5 mL.

After evaporation by nitrogen stream flow, a high-performance liquid chromatography (HPLC) system (Waters 2487, MILFORD, MA, USA) was used for the quantitative determination of SD concentrations during its transformation reactions. The HPLC system was equipped with a Waters 1525 binary pump and a dual λ absorbance UV/Vis detector (Waters 2487, UV–vis SPD-10AVP), with the wavelength used for detecting SD set at 254 nm. Samples of 20 μL were injected into the column through the sampling loop for analysis. SD separation was carried out using a Symmetry C18 column (5 μm beads, 250 × 4.6 mm I.D.) A mobile phase consisting of 75% methanol, 24% water, and 1% acetic acid (*v/v/v*) was applied at a flow rate of 1.0 mL/min.

Fresh leaf and root samples of lettuce plants collected from the control, 0.01, 0.05, 0.5, and 10 mg/L SD treatment groups were used to quantify antibiotic resistance genes. DNA extraction using the FastDNA[TM] SPIN Kit for Soil for root endophytes and leaf endophytes was performed using the FastPrep®-24 Instrument (both MP Biomedicals, Santa Ana, CA, USA) according to the manufacturer's instructions with minor modifications as described elsewhere [25]: Bead-beating was run twice for 45 s at a speed of 6.5 m s$^{-1}$. The samples were then centrifuged for 5 min at $14,000\times g$ and room temperature. Approximately 2 g of lettuce leaves or roots from each replicate was transferred into a 50 mL centrifuge tube, shaken at 200 rpm for 2 h after addition to 50 mL autoclaved 1 × phosphate buffer, and sonicated for 15 min. The washing solution was filtered with a 0.22 μm sterilized nylon net to collect the phyllosphere and rhizosphere microbiomes for DNA extraction using a DNeasy PowerWater Kit, Qiagen USA, Valencia, CA, USA. Primers developed for total bacteria, *sul1*, *sul2*, and *tnpA* genes and validated in previous studies were used in this study. Quantitative real-time PCR was performed on triplicate DNA extracts in independent runs for total bacteria, *sul1*, *sul2*, and *tnpA*. Each qPCR reaction was conducted using a Bio-Rad CFX96 Real-Time PCR Detection System (Bio-Rad, Herculers, CA, USA). The total reaction volume of 20 μL contained 1.0 μL DNA, 10.0 μL Talent qPCR mix (SYBR Green), and 0.5 μL of each primer (forward and reverse). The qPCR conditions for all of the genes consisted of an initial denaturation of 95 °C for 2 min, followed by 40 cycles of 15 s of denaturation at 95 °C, 15 s of annealing at the temperature specified in Table 1, and 40 s at 72 °C. A melt curve was run following each plate for primer specificity. The abundance of each gene in each sample was calculated by multiplying the number of copies per well by the total volume of DNA per well (1.0 μL). DNA standards were prepared from *E. coli*

strains carrying plasmids with total bacteria, *sul1*, *sul2*, and *tnpA* gene fragment inserts. Amplified DNA from SYBR Green assays was subjected to the melting curve analysis and gel electrophoresis to assure primer specificity. Samples of DNA were also selected from soil matrices for PCR product sequencing. DNA extracted from the soil was amplified with both forward and reverse primers (without SYBR green to prevent interference with the sequencing process), and the reaction products were purified using the E.Z.N.A.TM Gel Extraction Kit. The purified product subsamples were then submitted to the Biotechnology Company for sequencing.

**Table 1.** Primers employed in the present study for real-time quantitative PCR.

| Target Gene | Primer Pair a | Sequence (5′→3′) | Annealing Temp. (°C) | Reference |
|---|---|---|---|---|
| 16s rRNA | 16s rRNA-F 16s rRNA-R | GTGSTGCAYGGYTGTCGTCA ACGTCRTCCMCACCTTCCTC | 60 | [26] |
| *sul1* | sul1-F sul1-R | CGCACCGGAAACATCGCTGCAC TGAAGTTCCGCCGCAAGGCTCG | 62 | [27] |
| *sul2* | sul2-F sul2-R | CTCCGATGGAGGCCGGTAT GGGAATGCCATCTGCCTTGA | 60 | [28] |
| *tnpA* | tnpA-F tnpA-R | CCGATCACGGAAAGCTCAAG GGCTCGCATGACTTCGAATC | 60 | [29] |

The membrane permeability of roots was studied through electrolyte leakage (EL) measurements based on a procedure reported in previous studies. Briefly, 0.2 g of fresh roots was rinsed thoroughly with distilled water to remove surface contamination; then, the rinsed roots were cut into 1 cm segments and placed in individual vials containing 10 mL of distilled water. After exposure to vacuum treatment at 25 °C for 3 h, the electrical conductivity (EC) of the solution ($EC_1$) was measured using an electrical conductivity meter (SY-2, Institute of Soil Science, Chinese Academy Sciences, Nanjing, China). Samples were then placed in a thermostatic water bath at 100 °C for 15 min, and a second reading ($EC_2$) was determined after the solutions were cooled to room temperature. The EL values were calculated using the equation $EL = 100 \times EC_2/EC_1$.

Fresh leaf and root samples of lettuce collected from the control, 2 mg/L, and 10 mg/L SD treatment groups were used for the cell structure study via transmission electron microscopy, which was carried out on a Zeiss E.M.95 at 60 kV. Plant tissues were cut into 1 mm$^2$ slices, which were prefixed with 4% glutaraldehyde solution and then stored at 4 °C for 24 h after washing with 0.1 M sodium cacodylate buffer 3 times for 10 min each time. The samples were fixed in 1% osmium tetroxide at 4 °C for 2 h and then washed again with 0.1 M sodium cacodylate buffer 3 times for 10 min each. The samples were then dehydrated in acetone at 35, 50, 70, 95% (30 min each), and 100% (45 min each), 3 times. The dehydrated samples were infiltrated with 1/1 (*v/v*) resin and acetone for 2 h, 1/3 (*v/v*) resin and acetone overnight, and 100% resin overnight; then, each specimen was placed into beam capsules, filled with resin, and polymerized in an oven at 60 °C for 24 h. Then, 60- to 90-nanometer-thick sections were cut with a diamond knife on a Reichert ultramicrotome, stained for 10 min with uranyl acetate, and washed with 50% filtered alcohol and distilled water 2 times each. Sections were then stained with lead for 10 min and washed with double-distilled water.

The root length, dry matter yield, electrolyte leakage, and SD concentrations in plant organs were analyzed for deviations using Statistical Product and Service Solutions (SPSS 22.0). Duncan's multiple range test was used to compare the treatment means, and a 0.05 probability level was used to identify differences.

## 3. Results

### 3.1. Effect of SD Accumulation on Lettuce Growth

At all the studied SD concentrations, *Lactuca sativa* L. survived cultivation for 14 d, which indicated that this vegetable can grow under high SD concentrations, even up to 50 mg/L, in culture solutions. However, obvious morphological differences were found

among the lettuces under treatments with different concentrations of SD. At 0.01, 0.05, and 0.5 mg/L SD, both the roots and leaves exhibited little difference from those under the control treatments without SD in the culture solutions. With increasing SD concentration, phenomena of inhibited root and leaf growth were found, especially with SD concentrations higher than 10 mg/L, at which the lettuces showed an obvious decrease in biomass.

Accordingly, with increasing SD concentration in the culture solution, the dry matter yields in both the leaves and roots of the lettuce plants were reduced (Figure 1). There were differences in the dry mass among the SD treatments in the control, 0.01, 0.05, 0.5, and 2.0 mg/L groups, but the differences were nonsignificant. However, the total dry matter (roots + leaves) yields were significantly decreased in the treatments with SD concentrations of 10 and 50 mg/L. The mean dry mass values in each lettuce were 0.221, 0.22, 0.214, 0.22, and 0.19 g under the 0.01, 0.05, 0.5 and 2.0 mg/L SD treatments and the control, respectively. At higher SD concentrations, the dry mass decreased to 0.14 and 0.125 g under the SD treatments at concentrations of 10 and 50 mg/L, respectively.

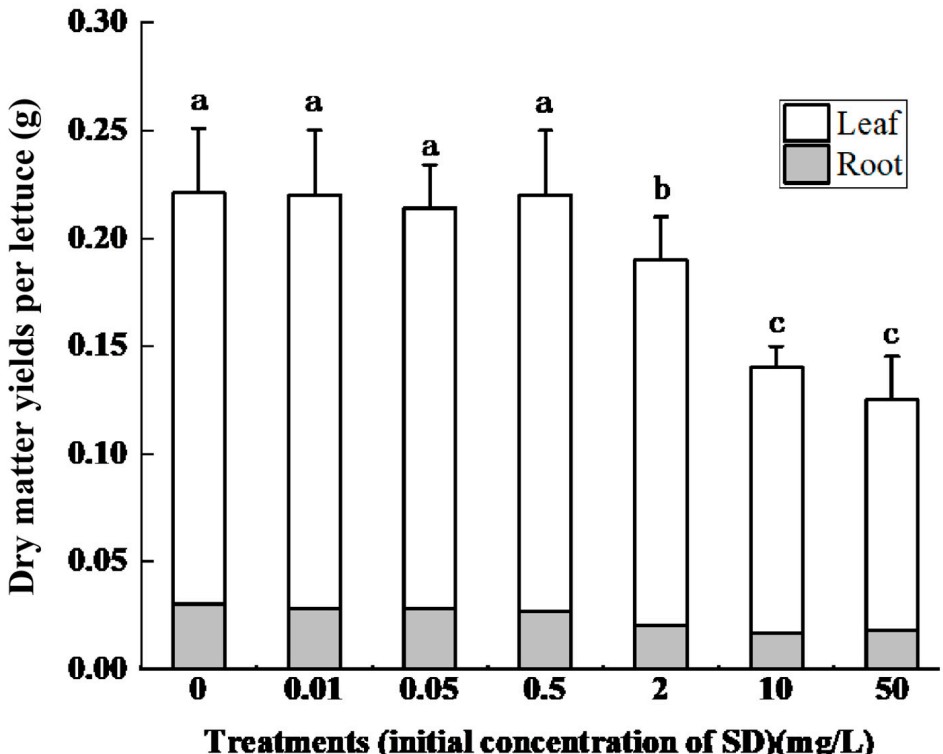

**Figure 1.** Effect of SD concentration on the dry matter yields per lettuce ($n = 10$). Error bars represent standard deviations of the means. a, b, and c on the top of the error bars represent significant differences ($p < 0.05$) among different initial concentrations of treatments.

The growth of roots is most sensitive to toxic effects and can act as a highly indicative parameter for evaluating the growth rate under SD stress environments; roots are the first and the primary organs through which plants contact the soil/aquatic environment, and they are also known to be the storage organs of nutrients, including toxic compounds [30,31]. The effects of different SD concentrations on the length of primary roots of lettuce on Days 0, 7, and 14 were studied (Figure 2). Without SD in the culture solution, the mean length of roots increased from 14.86 cm to 22.88 cm over the 14-day period, an increase of 1.5-fold. However, in the presence of SD, root growth was inhibited, and the inhibition rate was dependent on the SD concentration and exposure duration, as indicated by Figure 2. Up to Day 7, the root lengths grew to 3.63 cm in the control but only to 3.41, 3.24, 2.58, 2.38, 1.71, and 0.14 cm in SD stress environments with concentrations of 0.01, 0.05, 0.5, 2, 10, and 50 mg/L, respectively. Onward to Day 14, the root lengths under all SD treatments ($p < 0.05$) continuously increased,

but a large increase of 8.02 cm was observed in the control treatment, while under the 0.01, 0.05, 0.5, 2, 10, and 50 mg/L SD treatments, the increases in values compared with those on Day 7 were 7.78, 8.38, 5.37, 2.47, 1.87, and 0.03 cm, respectively. The above results clearly indicate the highly toxic effect of SD accumulation on the growth of lettuce roots and the whole dry mass, integrating the results of Figure 1.

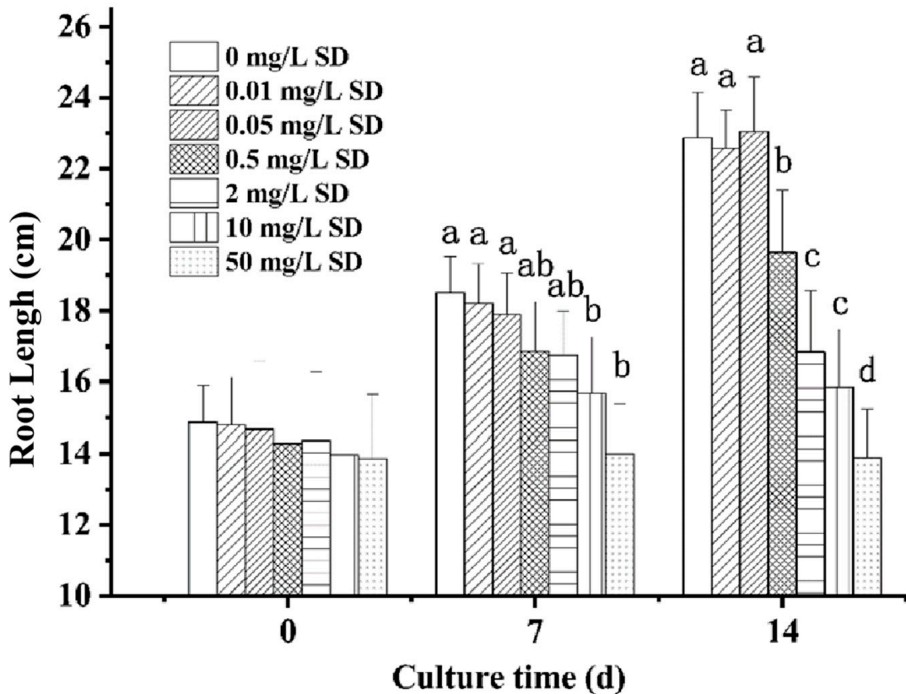

**Figure 2.** Effect of SD concentration on the root length of lettuce. Error bars represent the standard deviations of the means (*n* = 15), and a–d on top of the error bars represent significant differences (*p* < 0.05) among different initial concentrations of treatments.

### 3.2. Extent of SD Accumulation in Lettuce Organs

The morphological responses of lettuce to SD stress were attributed to the accumulation of SD in the organs of lettuce, which ultimately caused toxic effects on the plant. The accumulated concentrations of SD in roots and leaves of lettuce under different treatments are shown in Table 2. In the control experiments, no SD was detected in either the leaves or the roots. With SD in the culture nutrient solution, SD accumulated in the lettuce, including the roots and leaves; the accumulated SD concentrations in roots were higher than those in leaves under the four highest treatment concentrations (0.5, 2, 10, 50 mg/L) but not under the two lowest concentrations (0.01, 0.05 mg/L). In both roots and leaves, the accumulated SD concentrations significantly increased with increasing SD concentration in the culture nutrient solution. After 14 d of growth in the culture solution with 0.5 mg/L SD, the accumulated SD concentrations were 3.90 ± 0.91 and 6.48 ± 1.62 µg/kg in the leaves and roots, respectively. When the SD concentration in the solution was 2.0 mg/L, the accumulated SD concentrations increased by 2.4 and 3.5 times compared with those under 0.5 mg/L SD for leaves and roots, respectively. The increased accumulation rates of SD in roots were also found to be significantly higher than the corresponding increased rates in leaves under the same increased SD concentrations in the culture solutions.

**Table 2.** Concentrations (µg/kg in dry mass) of SD (±standard deviation) in the roots and leaves in different treatments.

| Treatment | Leaf | Root |
|---|---|---|
| Control | Not detectable | Not detectable |
| 0.01 mg/L SD | Not detectable | Not detectable |
| 0.05 mg/L SD | Not detectable | Not detectable |
| 0.5 mg/L SD | 3.90 ± 0.91 c | 6.48 ± 1.62 b |
| 2 mg/LSD | 9.55 ± 1.73 b | 22.86 ± 2.41 c |
| 10 mg/L SD | 13.06 ± 1.76 a | 39.80 ± 2.34 b |
| 50 mg/L SD | 16.74 ± 1.88 a | 120.87 ± 17.33 a |

a, b, and c within the same column represent significant differences ($p < 0.05$) among different initial concentrations of treatments ($n = 10$).

### 3.3. Antibiotic Resistomes in Plant Microbiomes

The roots and leaves of lettuce samples after 14 d of growth from the control and 0.01, 0.05, 0.5, and 10 mg/L SD treatment groups were collected to study the antibiotic resistance genes transformed in an SD stress culture environment. The absolute abundance levels of 16S rRNA in the lettuce samples after growth for 14 d under different treatments are shown in Figure 3. The absolute abundance levels of 16S rRNA in the phyllosphere and rhizosphere were significantly lower than those in the leaf endophytes and root endophytes, respectively. When SD was added to the culture solutions, the absolute abundance of 16S rRNA in all of the lettuce samples was reduced. The absolute abundance levels of 16S rRNA in the rhizosphere, root endophytes, and phyllosphere of lettuce under all SD treatments were significantly lower than those in the control group. In particular, the absolute abundance of 16S rRNA in the treatments with an SD concentration of 10 mg/L was significantly lower than that under the other SD concentrations.

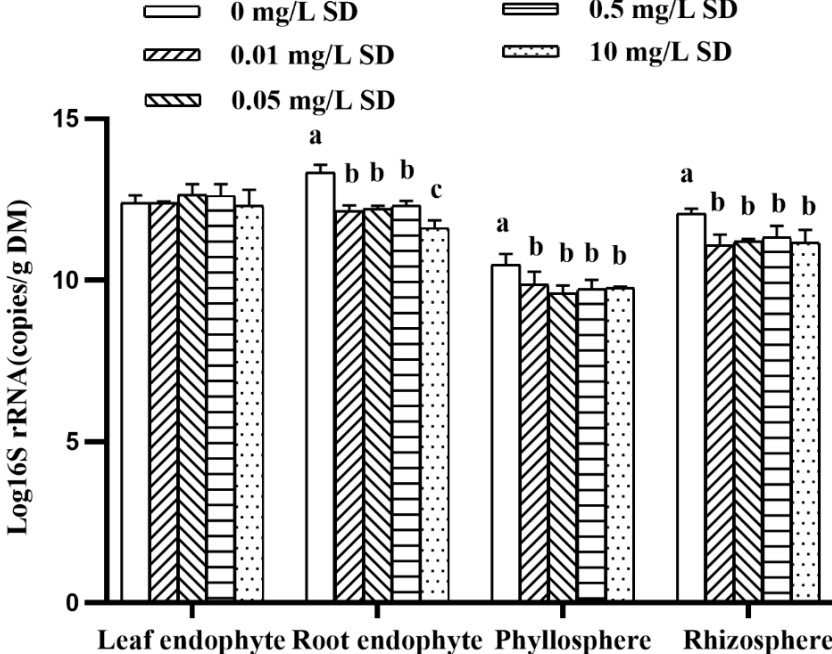

**Figure 3.** Effects of SD concentration on the log 16S rRNA absolute abundance in different parts of lettuce. Error bars represent standard deviations of the means. a, b, and c on top of the error bars represent significant differences ($p < 0.05$) among different initial concentrations of treatments.

Two common sulfonamide resistance genes, *sul*1 and *sul*2, and a transposon related to horizontal transfer, *tnpA,* were detected, and their relative abundance levels are shown in Figure 4. For *sul*1 and *sul*2, the relative abundance levels of root endophytes and leaf

endophytes were significantly lower than those of the rhizosphere and phyllosphere, and the leaf endophytes had the lowest abundance among all the samples. For both *sul*1 and *sul*2, the relative abundance of leaf endophytes was significantly lower than that of root endophytes. For the relative abundance levels of *sul*1 and *sul*2 in leaf endophytes, the abundance levels were significantly higher under SD concentrations of 10 mg/L than under other SD concentrations, while the *sul*2 relative abundance in the phyllosphere was similar.

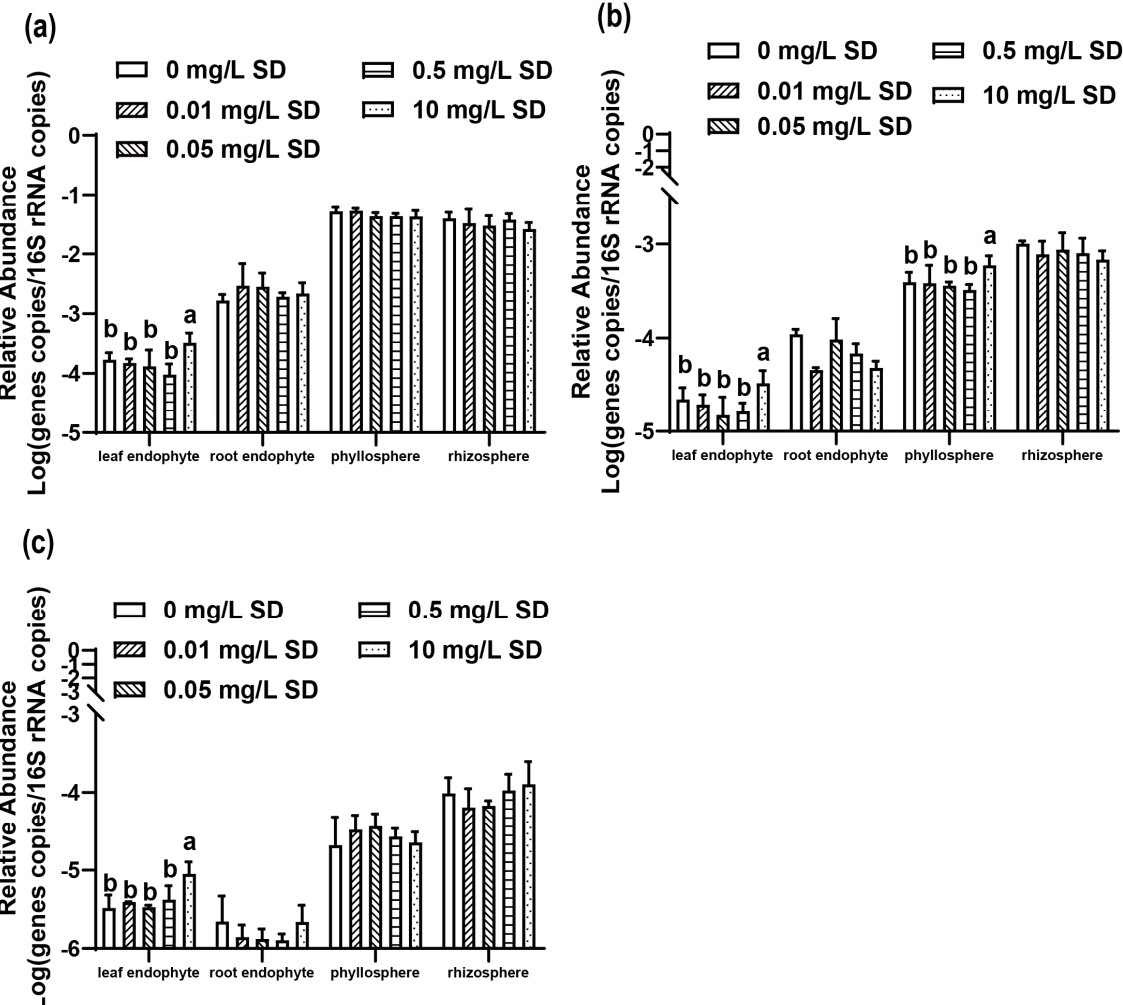

**Figure 4.** Effects of SD concentration on the relative abundance of antibiotic resistance genes (*sul*1, *sul*2) and *tnpA* in different parts of lettuce. Error bars represent standard deviations of the means. a and b on top of the error bars represent significant differences ($p < 0.05$) among different initial concentrations of treatments. (**a**) *sul*1; (**b**) *sul*2; (**c**) *tnpA*.

### 3.4. Membrane Permeability of the Roots

Electrolyte leakage (EL) has been suggested to be an indicator of membrane permeability and the degree of damage in roots [32]. The results of EL analyses of the lettuce roots after growth for 14 d under different treatments are shown in Figure 5. With the accumulation of SD in the roots during the lettuce growth processes in an SD stress environment, the leakage rates of roots were significantly increased. In the control group, the EL value of the roots was 20.0. Under SD stress environments at 0.5 and 2.0 mg/L SD, there were no significant differences compared with the control. When the SD concentration increased to 10 mg/L in the culture solution, the EL increased greatly, with an increase of 3.4-fold. When the SD concentration was further increased to 50 mg/L, the EL value increased to 79.0, with a significant difference obtained compared with that under 10 mg/L SD. The results also indicate that an SD concentration of 10 mg/L is the threshold for SD toxicity in

lettuce and can result in obvious physiological damage to the roots, ultimately leading to the inhibition of plant growth.

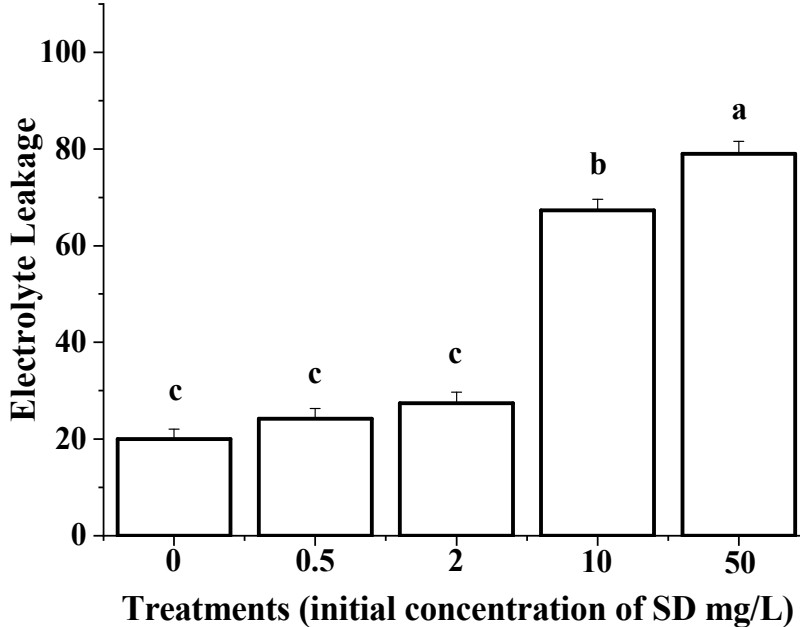

**Figure 5.** Effects of SD concentration on the electrolyte leakage of excised roots. Error bars represent standard deviations of the means. a, b, and c on top of the error bars represent significant differences ($p < 0.05$) among different initial concentrations of treatments.

### 3.5. The Ultrastructural Responses of Lettuce to the SD Stress Culture Environment

The roots and leaves of lettuce samples from the control, 2, and 10 mg/L SD treatment groups after 14 d of growth were collected to study their ultrastructural responses to the SD stress culture environment; the TEM results of root cells are shown in Figure 6a–c. The root cells of lettuce grown in the solution without SD showed a well-defined cell wall and regular cell contents for a normal lettuce root cell (Figure 6a), indicating no damage to lettuce under this condition. When the root cells of lettuce were grown with 2.0 mg/L SD, ultrastructural alterations, including thickening, were observed in the cell walls (Figure 6b). With a higher SD concentration of 10 mg/L, obvious damage and disintegration of cell walls were observed, including irregular cell walls and detached particles within the cell contents (Figure 6c).

Ultrastructural alterations were also observed in the leaf cells through TEM photographs of lettuce leaves collected from different treatment groups (Figure 6d–f). From the results, we observed that the chloroplasts of leaves from the control group were well elongated, having clear stoma and thylakoids with a typical grana structure and starch grains (Figure 6d). However, the chloroplasts in lettuce leaves under SD stress conditions at 2.0 mg/L showed damage and disintegration, although to a small extent, as indicated by Figure 6e. A further increase in SD concentrations in the culture nutrient solutions resulted in more obvious alterations of the chloroplasts. As shown in Figure 6f, the chloroplast structures in lettuce leaves under treatment with 10 mg/L SD, which was also observed as the threshold for SD toxicity in the study of membrane permeability, were severely altered in terms of the grana structure and starch grains, which appeared to be badly dissolved.

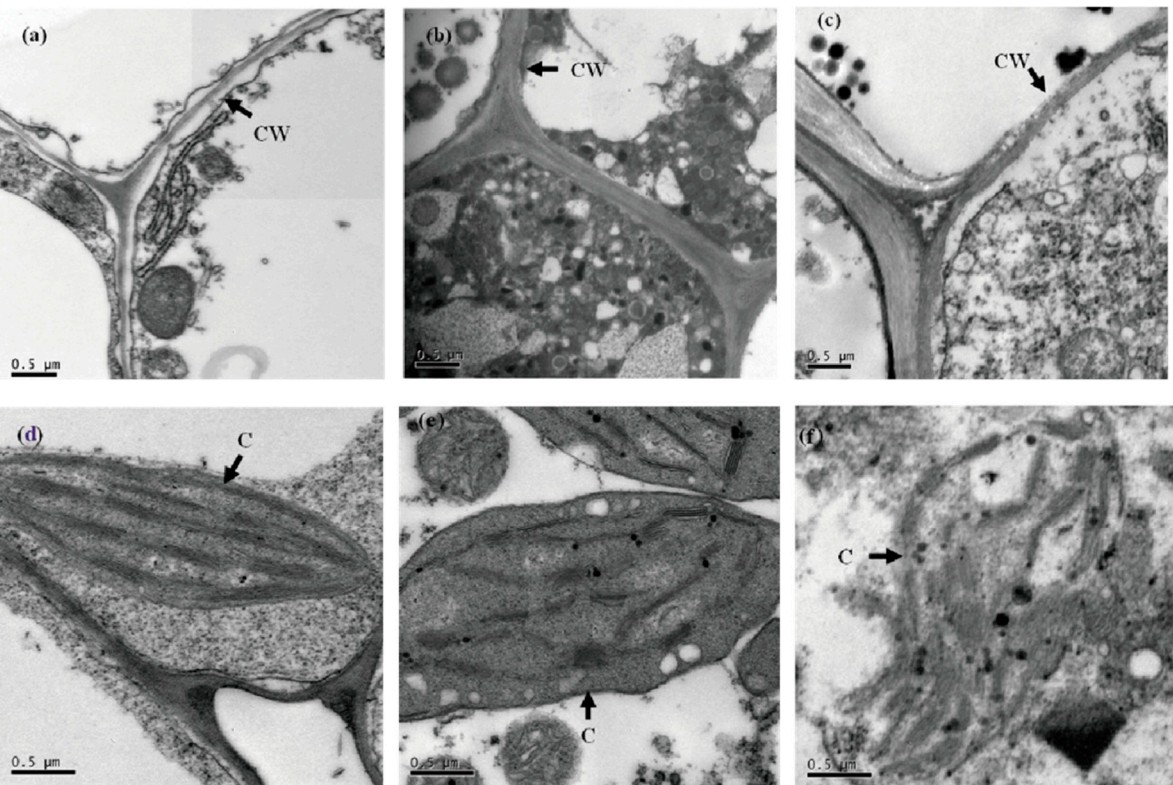

**Figure 6.** Transmission electron micrographs of lettuce root and leaf cells after 14 days of treatment under different SD concentrations: lettuce root under (**a**) control, (**b**) 2 mg/L, and (**c**) 10 mg/L; lettuce leaf under (**d**) control, (**e**) 2 mg/L, and (**f**) 10 mg/L. C denotes cell wall and CW denotes chloroplasts.

## 4. Discussion

Vegetable plants such as lettuce can absorb and accumulate residual SD antibiotics from the environment, which are ultimately enriched in the human body through direct consumption via the food chain. The USEPA reported that after being cultured in nutrient solution with 100 mg/L SD, the dry mass of *Lupinus albus* was reduced by 13% compared with that in the control [33]. In this study, *Lactuca sativa* L. was found to be more sensitive to SD than *Lupinus albus*: the dry mass yield reduction rates reached 36% with 10 mg/L SD in the culture solution. Our results also indicated that SD concentrations lower than 2.0 mg/L exerted no significant effect on the growth of lettuce, although the accumulation of SD and damage to the organs were still observed. Only at higher concentrations of SD, e.g., higher than 10 mg/L, did the morphology of lettuce, including the roots and leaves, exhibit significant changes in response to SD toxicity. Attributed to their direct and prolonged contact with SD residues, roots may be the main organs for accumulation and deposition in plants. As our results showed, the accumulated SD was mainly concentrated in the roots, with levels significantly higher than those in the leaves under all the different SD concentrations. On average, the SD concentrations in roots were two- to eightfold higher than those in leaves. Ahmed et al. also reported the bioaccumulation of six antibiotics in several plant species with obviously higher concentrations in the roots than in the stems [34]. The SD concentrations in *Lactuca sativa* L. are more related to the health of the human body since lettuces are usually directly eaten rather than processed.

Plant microbiota harbor intrinsic or acquired (from the environment) resistance genes that may be transmitted to humans through the food chain [16]. Environmental bacteria, especially rhizosphere bacteria, are an important type of plant endophytic bacteria [35], which can enter endophytic systems through plant tissues. Our results showed that the root and leaf endophyte microbiota were more abundant than the rhizosphere and phyllosphere microbiota. With the increase in SD concentration, the microbial abundance levels in the

root endophyte, rhizosphere, and leaf endophyte communities were significantly lower than those in the control group. This indicated that the addition of SD reduced the plant microbiota, which was also in good accordance with SD being a potent antibacterial agent. However, there was no significant change in the abundance of microbes within the leaf endophyte community because SD in lettuce is transported from roots to leaves and the amount of SD accumulation within the leaves is low.

*sul*1 and *sul*2 are the most common resistance genes to sulfonamides, so we selected them for this study. The abundance levels of these genes in both the rhizosphere and phyllosphere were significantly higher than those in both the root endophytes and leaf endophytes, which is consistent with previous reports on the transfer of antibiotic resistance from manure-amended soil to vegetables [36]. There are two potential explanations for the lower abundance of *sul*1 and *sul*2 in lettuce endophytes: (1) the endophyte community diversity is low [37]; and (2) there was no direct contact between endophytes and SD [38], which served as a selective pressure [36]. The relative abundance levels of *sul*1 in the leaf endophytes and in the rhizosphere and *sul*2 in the leaf endophytes were only significantly higher under the 10 mg/L SD treatments than those under the control treatments, and those under all other concentrations were not significantly different from the control levels. *sul*1 and *sul*2 were mainly concentrated in the roots and were significantly more abundant in roots than in the leaves under all the different SD concentrations, which is consistent with previous reports [36]. The relative abundance of *tnpA* was similar to that of *sul*1 and *sul*2 at different SD concentrations. In our study, SD was added to the nutrient solution as a pure chemical. The situation may differ from the conditions under actual agricultural practices, as they likely represent a mixed input of antibiotics, antibiotic-resistant bacteria, and antibiotic resistance genes.

The toxicity mechanisms of antibiotics accumulated in plants have been researched for a long time but remain unclear and, at times, even controversial. It has been reported that the phytotoxicity of enrofloxacin in *Lactuca sativa* plants generates both toxic effects and hormesis, which manifest as quickly accelerated growth; the toxic effect is reported to be induced by high concentrations of antibiotics (e.g., 5000 μg/L), while the hormetic effect occurs at relatively low concentrations (e.g., 50 and 100 μg/L) [39]. Liu et al. found that in Welsh onion leaves, chloroplasts are sensitive to antibiotics by promoting ROS accumulation [40], while other researchers suggested that antibiotics potentially inhibit plant growth because of hydrophobicity [41]. In the present study, our results from a cell structure study provide direct evidence to confirm that the chloroplasts in lettuce leaves are the structures most sensitive to SD accumulation that affects growth. The accumulation of a sufficiently high level of SD antibiotics in the leaf cell could damage the structure and function of chloroplasts, thereby affecting photosynthesis.

Based on the above results, the proposed possible mechanism of SD toxicity in lettuce is presented in Figure 7. The toxic effect of lettuce was expected to first occur in the roots due to the direct contact of this organ with SD during the growth process. This was also evidenced by the thickening of root cell walls in the antibiotic-stressed lettuce. The thickening phenomenon of the cell wall is believed to be a natural self-protection mechanism of the root cells to prevent further accumulation of the antibiotic from the rhizosphere and/or to prevent the enlargement of the cells due to the increased pressure built up within the cell contents [42]. However, under high SD stress with extremely high concentrations in the culture solutions, the root cells of the lettuce were still physiologically damaged (Figure 6c), which was expected to lead to an equilibrium in the SD distribution between the external solution and the sap of the xylem. The physiological damage to the root cells was also reflected by the drastic increase in the EL rates obtained in the roots from lettuce grown under treatments with high SD concentrations.

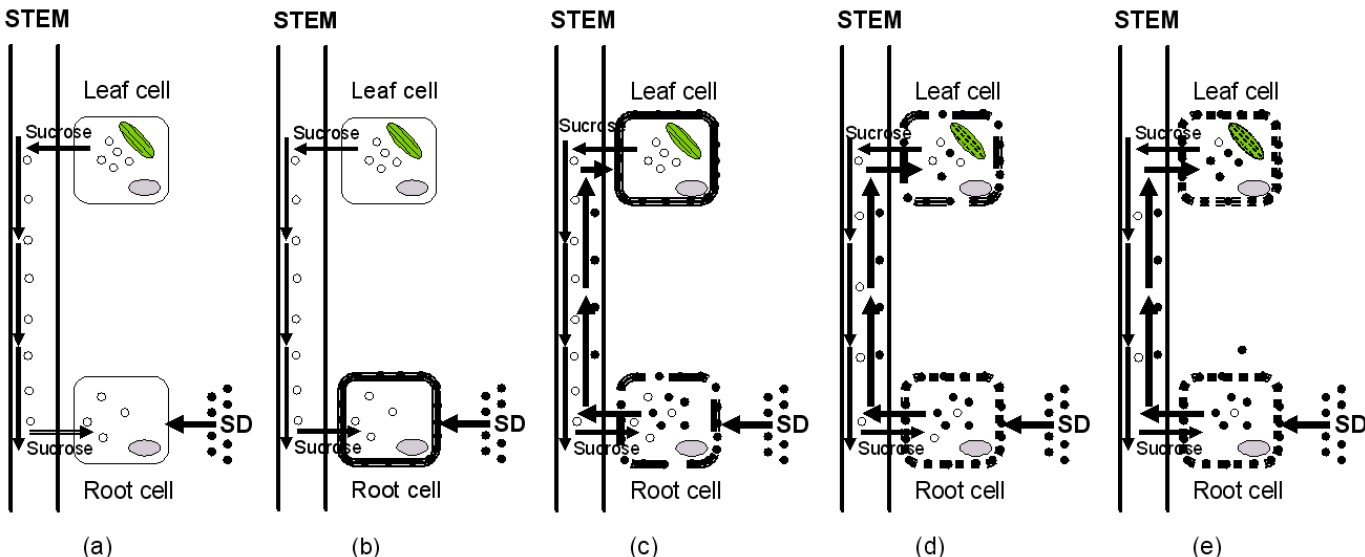

**Figure 7.** Proposed toxicity mechanism of accumulated SD in lettuces grown in SD stressed culture solutions. (**a**) lettuce at first contact with SD; (**b**) thickening of the lettuce root cell in the SD-stressed; (**c**) lettuce root cell damage and SD translocated from the roots to the leaves; (**d**) lettuce leaf cell damage and SD damage to the chloroplasts; (**e**) lettuce leaf cell broken, thus ultimately inhibiting plant growth.

After entering the xylem, the antibiotic can be easily translocated from the roots to the leaves along with the transpiration stream, resulting in accumulation of the antibiotic in the leaves and leading to further damage to the leaf cells, especially the chloroplasts. Chloroplasts are the organelles directly responsible for photosynthesis, and after being destroyed by the toxic effects of antibiotics, they lose the important ability to produce chlorophyll for the conversion of solar energy into chemical/food energy in the plant, thus ultimately inhibiting plant growth.

## 5. Conclusions

In conclusion, this study presents a possible mechanism of SD accumulation in lettuces grown in SD stressed culture solutions and antibiotic resistance gene transformation. The mechanism we have proposed is unverified, so SD may inhibit lettuce growth via other ways. Our study provides valuable information on the uptake and accumulation of antibiotics and antibiotic resistance genes in plants, which is useful for environmental fate and risk assessments of antibiotics for human exposure.

**Author Contributions:** Conceptualization, Y.W.; project administration, Y.W., L.M.; data curation, L.M., Y.-X.C.; performed the experiments, Y.W., L.M., Y.-X.C., C.W., J.-H.C., Z.-J.Z., M.-Y.Z., J.-T.F.; writing—review and editing, L.M., Y.W. All authors have read and agreed to the published version of the manuscript.

**Funding:** This study was supported by This study was supported by Guangdong agricultural research projects (YUECAINONG202137).

**Institutional Review Board Statement:** Not applicable.

**Informed Consent Statement:** Not applicable.

**Data Availability Statement:** Data are contained within the article.

**Acknowledgments:** We acknowledge the support in terms of time and facilities from South China Agricultural University for this study.

**Conflicts of Interest:** The authors declare no conflict of interest.

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
