# Peer review of "Antibiotic Resistance Gene Transformation and Ultrastructural Alterations of Lettuce (Lactuca sativa L.) Resulting from Sulfadiazine Accumulation in Culture Solution"

_processes, doi:10.3390/pr9081451_

Round 1

Reviewer 1 Report

The manuscript describes the impact, at several levels, of an antibiotic (sulfadiazine) on lettuce. the manuscript and subject is really interesting and of extreme importance in our current situation. The authors have used several techniques to see the impact of the antibiotic on the vegetable and bacterial communities. The main issue is really the english, that must be improved prior to publication and also the lack of description of the PCR conditions, including the primers and probes used (with respective references), the pcr conditions (master mix, primers and probes concentration, etc), and the cycling conditions.

minor comments:

Abstract

  • line 24: I believe the authors mean lettuce rather than lecture;

Introduction

  • page 2, line 49: separate from from 1.93;
  • page 2, lines 52-53: what do the authors mean with 'endpoint'?
  • page 2, lines 56-57: references are missing;
  • page 2, line 64: substitute As by arsenic;
  • page 2, line 68: Arabidopsis thaliana in italic. the same is valid throughout the text for this plant and others that are described throughout the text;
  • page 2, line 79: substitute antibiotic resistant gene for antibiotic resistance gene (check other places where this applies

Materials and methods:

  • page 3, line 105: temperature of incubation?
  • page 3, line 116: substitute 'with little revision' with 'minor modifications'
  • page 3, line 136: the authors state "with minor modifications as described elsewhere". what were the modifications and reference;
    1.  

  • as said before, little information is provided on the pcr part of the materials and methods, and this should be further described, including primers and probes used, mastermix conditions and cycling conditions;
  • the concentrations used for the standards of the pcr;
  • page 4, line 164: do the authors really mean mm3?

Results

  • page 5, line 215: change P with p
  • page 6, line 236: do the authors mean increased when they have creased?
  • page 8, line 275: careful with the reference format for this journal
  • page 10, figure 6: what is the meaning of the letters on the figures. CW, C? describe in the legend

Discussion

  • page 11, lines 352, 362: always write the name of the gene in lower caps (sul1);
  • page 11, line 368: write in full what each abbreviation mean. as this was not done before.

Reviewer 2 Report

 Antibiotic resistant gene transformed and Ultrastructural Alter-2 nated of Lettuce (Lactuca sativa L.) from Sulfadiazine Accumulation Solution

The authors propose a study exploring the effects of the antibiotic sulfadiazine on growth and ultrastructural changes on Lactuca sativa, and on the expression of antibiotic resistance gene in microbiomes. The study is interesting and properly conducted, but manuscript needs a major revision of English and style. The graphs are of poor quality.

Abstract

            Line 24 please correct “lecture”

Introduction

            References 1 and 2 are not appropriate to support the first statement

Results

            Fig. 1: How many lettuces were analyzed per group? Please specify (the same for Table 1)

Lines 215-217:            with 0.5, 2, 10 and 50 mg/L SD treatments, the increased values compared that on day 7 were 8.01, 7.78, 5.37, 2.47, 1.87, and 0.03 cm respectively. You mentioned 6 different values for 4 concentrations. Please correct

Fig. 2: please amend Y axis label

The breaks and changes in Y axis in fig 4 make it difficult to read. Log scale might help, or find another way to represent your data. 

Figure 5, legend: “a, b, c, d, and e indicated the different differences among treatments with different SD concentrations”. E is absent in figure, d is different against what? Do letters a-e represent different levels of significance? What about p? (please specify in every figure)

Figure 6: please add the legend for letters on pictures. It might help the comparison between leaves and roots in the different conditions to put the corresponding images side by side (or in the same column), for instance

a

b

c

d

e

f

Discussion

            Lines 337-338 the most edible part of Lactuca sativa are leaves, not roots.

Line 368. Please make ABs, ARB, and ARGs explicit.

Author declare aiming at disclosing SD toxicity mechanism, but with the results produced in this study only a hypothesis can be formulated, therefore please amend the aim of the study.

The conclusion session is a summary of the results. Please rewrite proper conclusions, include the limits and strength of the study, and the value and significance/perspectives of the study

Round 2

Reviewer 2 Report

The manuscript has been greatly improved, but still it is not clear what "a, b, c and d" mean in the different figures (cfr legends of figures 1, 3-5 and table 2. In figure 2 authors use asterisks to identify significant differences). Do they correspond to different p? If yes, please specify.

Author Response

Point 1: The manuscript has been greatly improved, but still it is not clear what "a, b, c and d" mean in the different figures (cfr legends of figures 1, 3-5 and table 2. In figure 2 authors use asterisks to identify significant differences). Do they correspond to different p? If yes, please specify.

Response 1: Thanks for your comment. We have change the legends of figures 1-5 and table 2 in the revised manuscript(Line 246, 271, 295, 313, 331 and 354).